# Tryptamine accumulation caused by deletion of *MrMao-1* in *Metarhizium* genome significantly enhances insecticidal virulence

Xiwen Tong[1,2☯], Yundan Wang[1☯], Pengcheng Yang[3], Chengshu Wang[4], Le Kang[1,2,3]*

1 State Key Laboratory of Integrated Management of Pest Insects and Rodents, Institute of Zoology, Chinese Academy of Sciences, Beijing, China, 2 University of Chinese Academy of Sciences, Beijing, China, 3 Beijing Institutes of Life Science, Chinese Academy of Sciences, Beijing, China, 4 Key Laboratory of Insect Developmental and Evolutionary Biology, Shanghai Institute of Plant Physiology and Ecology, Chinese Academy of Sciences, Shanghai, China

☯ These authors contributed equally to this work.
* lkang@ioz.ac.cn

**Data Availability Statement:** All sequencing files are available from the SRA database (accession number: SRP145755.)

## Abstract

*Metarhizium* is a group of insect-pathogenic fungi that can produce insecticidal metabolites, such as destruxins. Interestingly, the acridid-specific fungus *Metarhizium acridum* (MAC) can kill locusts faster than the generalist fungus *Metarhizium robertsii* (MAA) even without destruxin. However, the underlying mechanisms of different pathogenesis between host-generalist and host-specialist fungi remain unknown. This study compared transcriptomes and metabolite profiles to analyze the difference in responsiveness of locusts to MAA and MAC infections. Results confirmed that the detoxification and tryptamine catabolic pathways were significantly enriched in locusts after MAC infection compared with MAA infection and that high levels of tryptamine could kill locusts. Furthermore, tryptamine was found to be capable of activating the aryl hydrocarbon receptor of locusts (*LmAhR*) to produce damaging effects by inducing reactive oxygen species production and immune suppression. Therefore, reducing *LmAhR* expression by RNAi or inhibitor (SR1) attenuates the lethal effects of tryptamine on locusts. In addition, MAA, not MAC, possessed the monoamine oxidase (*Mao*) genes in tryptamine catabolism. Hence, deleting *MrMao-1* could increase the virulence of generalist MAA on locusts and other insects. Therefore, our study provides a rather feasible way to design novel mycoinsecticides by deleting a gene instead of introducing any exogenous gene or domain.

## Author summary

Mycoinsecticides are widely used in place of chemical pesticides to protect crops from pest damage. *Metarhizium* spp. fungi specifically live inside the body cavity of insects and can produce insecticidal metabolites, such as beauvericin, destruxins, and taxol. The variable virulence between host-generalist fungus *Metarhizium robertsii* (MAA) and host-specialist fungus *Metarhizium acridum* (MAC) to locusts was studied. We found that MAC is

**Funding:** Prof. Le Kang received the Strategic Priority Research Program of the Chinese Academy of Sciences, Grant No. XDB11010000 (http://www.cas.cn/xxgkml/zgkxyyb/kxyj/xdzx/). Dr. Yundan Wang received National Natural Science Foundation of China, Grant No. 31670420 (https://isisn.nsfc.gov.cn/egrantweb/). The funders had no role in study design, data collection and analysis, decision to publish, or preparation of the manuscript.

**Competing interests:** No authors have competing interests.

more virulent than MAA on locusts, and MAC-infected locusts display higher amounts of tryptamine than do MAA-infected locusts. Furthermore, MAC cannot produce destruxins, but can produce abundant tryptamine to kill locusts when accumulated owing to the absence of a gene for tryptamine catabolism in the MAC genome. Tryptamine activates the aryl hydrocarbon receptor of locusts (*LmAhR*) that produces damaging effects by regulating reactive oxygen species production and suppressing the immune system of locusts. Therefore, the deletion of *MrMao-1* in the generalist fungus MAA significantly improves the virulence of the fungus to locusts and other insect species. The resulting new insights into the core metabolism of high virulence of host-specialist fungus can provide an improved basis for designing mycoinsecticide strains.

## Introduction

Pest insects have been the important threat to agricultural production and health of animals and humans in the world. The potential of entomopathogens as biological control agents of pest insects is widely recognized because the biological control is a feasible alternative to chemical insecticides in the management of insect infestations. Entomopathogenic fungi as biological control products are widely applied to control the populations of various insects [1,2] because of the favorable properties of fungus, such as lack of pesticide residue and their safety for humans and the environment. *Metarhizium* spp. are the most common insect-pathogenic fungi or mycopesticides that control many species of insects [3]. Unlike other microbial pesticides that infect insects through the gut, such as microsporidia, bacteria, and viruses, mycopesticides infect insects by directly penetrating the cuticle. Once inside the insect's cuticle, *Metarhizium* spp. will make their way to the hemolymph, where their differentiate into blastospores that produce insecticidal metabolites such as destruxins [4,5], resulting in insect death within several days [6]. Recent efforts have focused on improving fungal virulence against their insect hosts to make mycopesticides more efficient. Thus, the efficacy of mycopesticides is one of the most important competitive factors being compared with traditional chemical pesticides in pest control.

As a biocontrol agent of insect pests, the generalist MAA has a broad range of hosts, but shows lower virulence to grasshoppers and locusts [7]. By contrast, the specialist MAC effectively kills grasshoppers and locusts rather than other non-target insects. Previous research proved that the formation of appressoria by *Metarhizium* germlings determined the successful infection of hosts [8]. The esterase gene (*Mest1*) involved in cuticular penetration in MAA enabled MAC to successfully infect caterpillars [9]. Thus, host specificity of *Metarhizium* probably relies on the cuticular penetration stages. *Metarhizium* spp. have approximately 15% of putative genes associated with virulence in the total proteins encoded in their genome [7,10]. Further comparative genomic analysis showed that MAC presents less virulence genes, including chitinase, protease, and secondary metabolic gene clusters [7]. Interestingly, MAC lacks the genes for the biosynthesis of the virulence factor destruxin compared with MAA [7,11]. In fact, destruxin allows MAA to inhibit the system activity of prophenoloxidase and the production of bactericidal peptides of hosts, suppressing the hosts' immune defense and increasing the opportunistic infection of microbes in host insects [12,13]. MAC lacks enzyme genes required for the catalysis or synthesis of molecules because of the simplicity of its genome. This loss of genes causes either the absence of virulence factors or the accumulation of other harmful chemicals in insects during MAC infection. The higher virulence of MAC to

grasshoppers and locusts implies that MAC probably has a supply of other chemicals that can kill locusts. However, the virulence factors of MAC remain unknown.

*Metarhizium* species are fungal insect pathogens that can synthesize secondary metabolites of insecticidal and antibacterial compounds [11,14]. The metabolite destruxin is considered the most important insecticidal factor of the *Metarhizium* species. Destruxin specifically suppresses the host production of anti-bacterial AMPs (antimicrobial peptides) without affecting anti-fungus AMPs, whereas, the existence of bacteria induced the suppression effects of destruxin on anti-fungus AMPs [15], leading to increase the opportunistic infection of bacteria to defeat host defense for fungus growth. Our previous study demonstrated, during infection of locusts with MAA, that the gregarious and solitary locusts displayed very different immune responses, gregarious locusts have stronger tolerance to MAA through up-regulation of *gnbp* and inhibitors of ROS production [16]. Given that MAC cannot produce destruxin, other metabolites may cause it to have high virulence against grasshoppers and locusts. Because the specific horizontal gene transfer between MAA and MAC from plant, bacteria, fungus and archaea [10], we noticed that plants use indole alkaloids to resist insects [17,18], and synthesized indole derivatives have high virulence against insects [19,20]. Interestingly, indole acetic acid (IAA) exerts its virulent effect by co-infection with fungal spores into the hemocoel of insects, possibly activating the prophenoloxidase cascade of insects and inducing the production of ROS [21]. The tryptophan metabolism pathways in fungi can produce many indole derivatives [22]. However, specific insecticidal chemicals and the insecticidal mechanism of MAC remain unclear.

In this study, we conducted a series of comparative studies between specialist and generalist fungal pathogens (MAA and MAC, respectively) against the migratory locust *Locusta migratoria*. The transcriptome analysis revealed that the specialist fungal pathogen affected the expression of detoxification genes and immune response genes, whereas the generalist fungal pathogen did not affect these genes in locusts. A significantly higher level of tryptamine was accumulated in the hemolymph of locusts infected by MAC than in those infected by MAA. Importantly, the virulence difference between the two *Metarhizium* species was highly associated with the evolutionary absence of multiple copies of tryptamine catabolic enzyme genes (*Mao*) in the genomes of MAC. The mutant strain displayed higher virulence to locust and other insect species caused by deleting the target gene *MrMao-1* in generalist MAA. Our data unraveled a previously unsuspected fungal virulence factor on the basis of the genome evolutionary clues of the fungus, suggesting a new pathway to develop highly efficient mycoinsecticides without any introduction of exogenous gene.

## Results

### Differential responses of locusts against MAA and MAC infection

First, virulence bioassays of the acridid-specific MAC and the generalist MAA were performed via topical infection under the pronotum of locusts with $2 \times 10^3$ conidia of fungi. The median lethality time ($LT_{50}$) of the locusts infected by MAC was 2 days shorter ($P < 0.01$, $LT_{50} = 5.55 \pm 0.38$ days) than those infected by MAA ($LT_{50} = 7.33 \pm 0.45$ days) (Fig 1A). Thus, the host-specialist MAC killed locusts more quickly than did the host-generalist MAA.

RNA-seq transcriptome was conducted to investigate the immune responses of locusts to the two species of fungal pathogens by analysis of the fat bodies of the locusts after MAC and MAA infection at their respective $LT_{50}$. The results showed that 359 genes were jointly induced in the locusts infected by MAC and MAA compared with the mock control. Approximately 800 genes were uniquely triggered by MAC, whereas only 110 unique genes were activated in

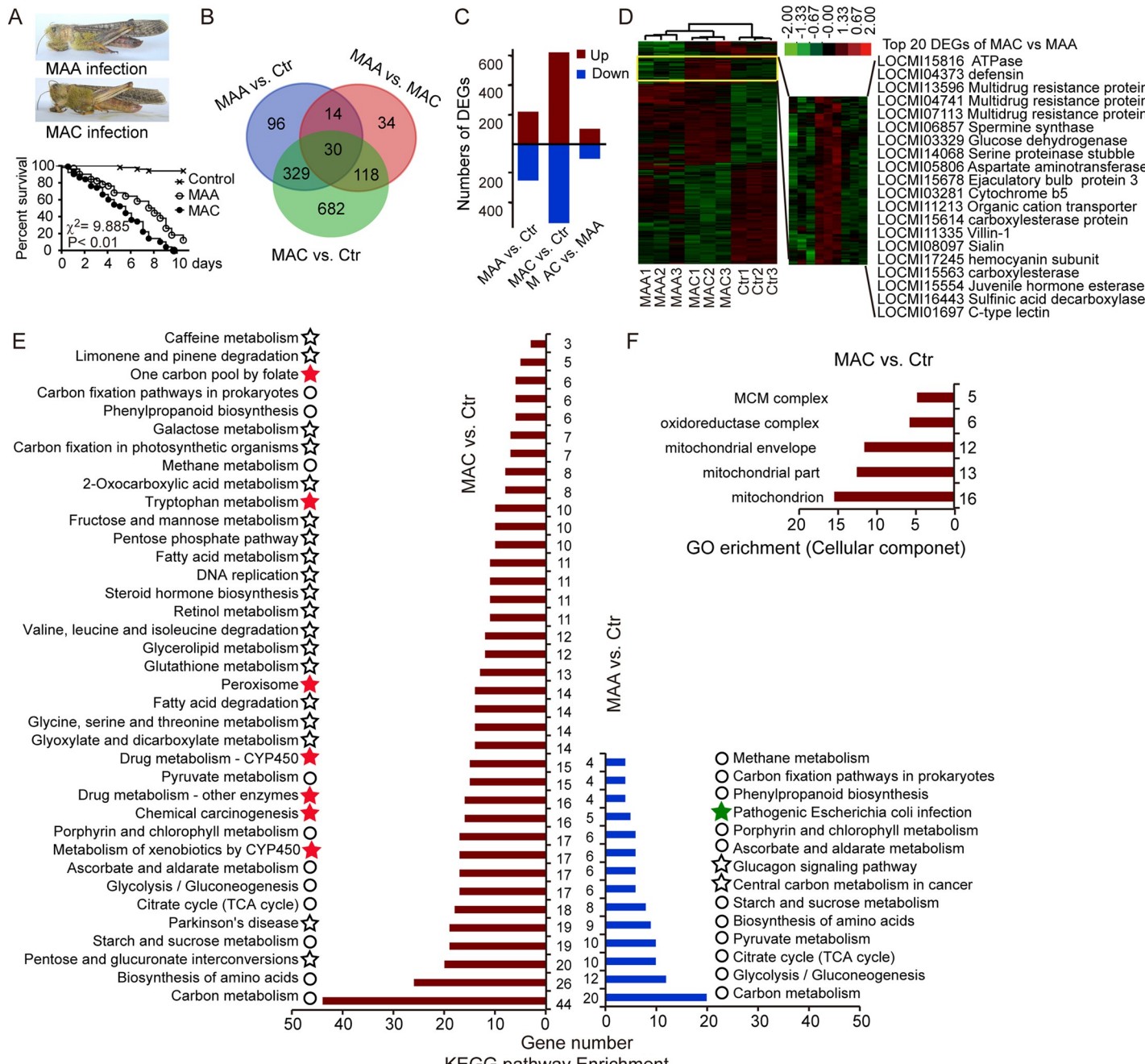

**Fig 1. Generalist MAA and specialist MAC fungus induced different responsiveness of locusts.** (A) Mycosis and survival of locusts after infection by MAA and MAC. Adult male locusts were used in the bioassay and estimation of $LT_{50}$ values. A significant difference was observed between the locusts infected by these two fungal species after Kaplan–Meier analysis ($n_{control}$ = 49, $n_{MAA}$ = 50, $n_{MAC}$ = 50, P < 0.01). (B) Venn diagram results showing differentially expressed genes in fat body samples infected by MAA and MAC after transcriptome analysis of locusts' fat body samples. (C) Transcriptome analysis showing the number of differentially up- and downregulated genes of each treatment for comparison. (D) Heat map of differentially expressed genes in locusts showing the distinct pattern of MAA and MAC infection. The right panel shows the genes of locusts distinctively induced by MAC infection. The colored bar indicates the normalized gene expression by z-score transformation.(E) After the enrichment of differentially expressed genes, MAA uniquely induced three pathways of infected locusts (labeled as star figure, green color indicates the immune response), whereas MAC uniquely induced 26 unique pathways of infected locusts (labeled as star, red star indicates the detoxification pathways, including xenobiotic response and potential toxics catabolism; circles indicate the pathways in fat body cells of locusts induced by either MAA or MAC infection). (F) GO enrichment showing the unique class (cell component), including mitochondrial part and oxidoreductase induced by MAC infection.

response to MAA infection. By directly comparing the locusts infected by MAA and MAC, we found that 162 genes (sum of 118, 30, and 14) involved in response to fungal infection (Fig 1B). Among these differentially expressed genes (P < 0.01; FDR < 0.05), MAC infection upregulated 621 genes, while MAA infection upregulated 221 genes. A similar number (ca. 100) of genes was up- and down-regulated in the fat body of the locusts by direct comparison of the infections by MAC and MAA (Fig 1C). Noticeably, clustering analysis of DEGs indicated that detoxification genes (e.g., multidrug-resistant proteins) and immune defense genes (e.g., *cactus*, *lectin*, and *serine proteases*) were specifically up-regulated by MAC infection (Fig 1D and S1 Table). Interestingly, MAC induced higher level of *defensin* than MAA. However, KEGG enrichment showed that MAC infection induced unique pathways in locust fat bodies were detoxification pathways, including xenobiotic response and chemical catabolism. Among the chemical metabolism pathways, tryptophan metabolism provided the cues to discover the potential virulence factors of secondary metabolites in MAC. On the contrary, MAA infection induced the classic immune pathway (green star labeled) (Fig 1E). Enrichment analysis of gene ontology (GO) showed that oxidation–reduction was the significant ontology triggered by the MAA and MAC infections (S1A and S1B Fig). Importantly, GO enrichment showed that only MAC infection induced the unique class of cell components, including oxidoreductase complex and mitochondrial component (Fig 1F). Thus, locusts mostly relied on the detoxification pathways to resist MAC infection compared to MAA infection.

## MAC infection induced distinct detoxification genes

Analysis of quantitative real-time polymerase chain reaction (qRT-PCR) showed that mRNA of cytochromes P450 (CYPs) in enriched detoxification pathways were expressed at significantly higher levels in locusts infected by MAC than in those by MAA (Fig 2A). To understand the chemicals for inducing detoxification, we investigated the genes in tryptophan catabolism, including two putative *LmMao* genes (NCBI accession numbers: MK125030 and MK125031; contributing to the conversion of tryptamine to indole-3-acetaldehyde) and a putative tryptamine hydroxylase (NCBI accession number: MK125032; CYP6J1 family protein) and found that they were highly expressed (P < 0.01) in locusts infected by MAC compared to MAA (Fig 2B). Interestingly, we found that the expression of putative tryptophan decarboxylase (TDC), *LmTdc* (NCBI accession number: MK140601), similar to the *MrTdc* of MAA (40% identity at the amino acid level) that produces tryptamine, was not affected in locusts after infection by both *Metarhizium* species (Fig 2C). These results suggested that locust did not alter the tryptamine production in respond to fungus infection. To understand the high level of *LmMao* genes expression for tryptamine catabolism in respond to MAC infection, we considered that MAC may produce the tryptamine. By comparing the genomes of MAA and MAC, it was found that the core enzyme for tryptamine catabolism pathway was absent in MAC (Fig 2D). These results suggested that locusts deployed the detoxification process for tryptamine catabolism as the priority resistance against MAC infection.

## Tryptamine showed toxicity to locusts by inducing ROS stress

The toxicity of tryptamine to locusts was verified by first testing its median lethal dosage ($LD_{50}$), which was determined to be between 1.585–20.795 (ng/g) at 95% confidence interval, in locusts (Fig 3A and S2 Table). This lethal dosage confirmed the insecticidal efficiency of tryptamine on locusts. Cellular reactive oxygen species (ROS) production was clearly observed in locusts 4 h after the tryptamine injection (10 ng per insect) (Fig 3B). Tryptamine injection also induced the upregulation of locust *LmMao-a* and *LmMao-b* genes (Fig 3C). Moreover, the detoxification genes (*cyp4* and *cyp6*) and immune response genes (*cactus*, *stubble*, and *easter*)

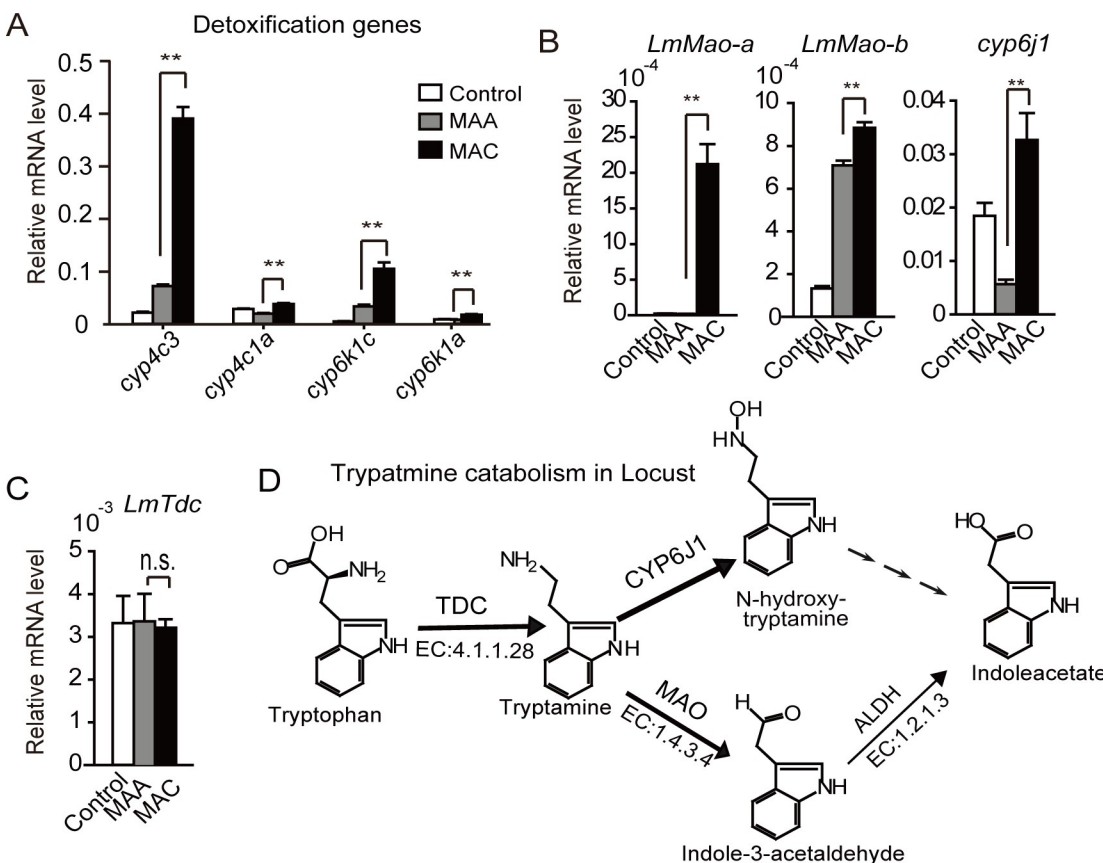

**Fig 2. Locust preferred detoxification genes and tryptamine catabolism genes to resist MAC infection.** (A) The detoxification genes of locusts were verified by qRT-PCR analysis. (B) qRT-PCR analyzed the gene expression of tryptamine catabolic enzyme in locusts after fungal infections. The results indicated that the gene expression of tryptamine *Mao* (*LmMao-a* and *LmMao-b*) and tryptamine hydroxylase *cyp6j1* significantly increased in the fat bodies of MAC-infected locusts. However, (C) To observe the endogenous production of tryptamine of locust self, we performed the qRT-PCR analysis and found that the expression of the *LmTdc* gene for tryptamine production displayed no significant difference between pre- or post-infection of MAC. Thus, we speculate that the increase of tryptamine production mainly results from the fungus.(D) Given that KEGG enrichment showed that the detoxification pathways and tryptophan metabolism pathway of locusts was induced by MAC infection, we analyzed the tryptophan catabolism pathways in locusts. The arrow width indicates the main enzymes in insect tryptophan catabolism. All qPCR data were presented as mean ± standard error of the mean (SEM), Student's t-test. ** indicated P < 0.01. n.s. indicated no significance.

were upregulated by tryptamine injection. Interestingly, the representative antimicrobial gene *defensin* was suppressed (Fig 3C).

   Tryptamine receptor AhR in insects is translocated from the cytoplasm into the nucleus to activate its function as a transcription factor that regulates gene expression [23]. Hence, we performed an immunoblot analysis, which showed that tryptamine injection induced the translocation of AhR in the fat body cells of locusts (NCBI accession No: MH393891) (Fig 3D). Fig 3E shows that a high level of ROS in locusts induced by tryptamine was repressed after co-injection with StemReginin 1 (SR1), a tryptamine antagonist. Moreover, SR1 reduced the effects of tryptamine on the expression of detoxification and immune defense genes (Fig 3F). Injecting double-stranded RNA (dsRNA) of AhR also effectively decreased AhR expression and ROS level induced by tryptamine (Fig 3G and 3H). The reduction of AhR expression by RNAi also showed similar efficiency with the AhR antagonist SR1 (Fig 3I). These results indicated that ROS induced by tryptamine was mediated by AhR. Immunoblot analysis showed

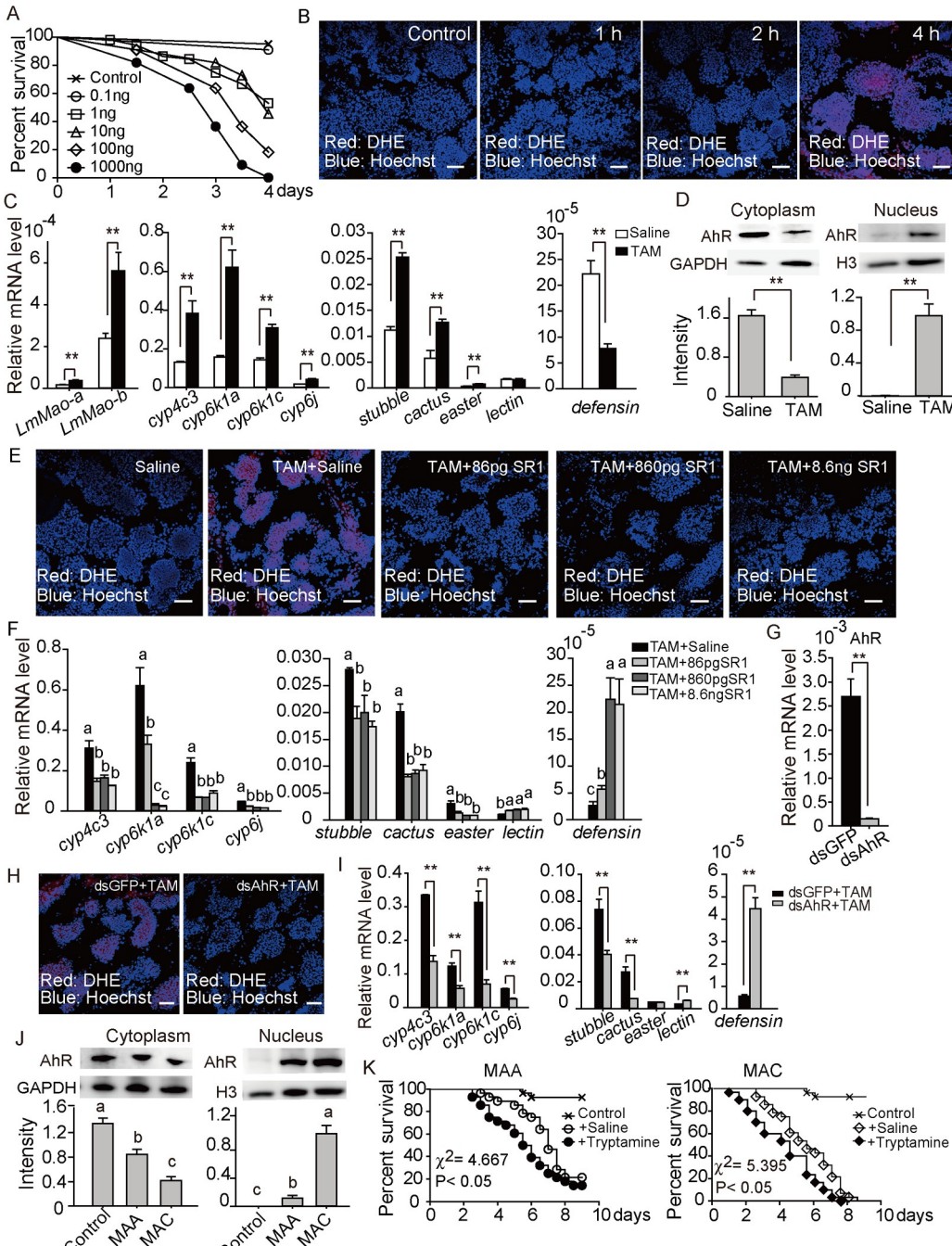

**Fig 3. Tryptamine showed high toxicity to locusts.** (A) The locusts' survival after the injection of different tryptamine dosages (0.1, 1, 10, 100, and 1000 ng/insect) showed the median lethal dose and sub-lethal dose. Data were calculated by using Kaplan–Meier methods and Probit methods in SPSS 20.0. (B) Fluorescent confocal microscopy imaging showed that the fungus with higher levels of tryptamine induced more ROS production in locust fat body cells. Red cells indicated that DHE staining the DNA of cells was degraded by ROS. Blue cells indicated Hoechst 33342 staining. Four hours after 10 ng tryptamine injection, ROS production was remarkably presented by observing DHE staining. (C) qPCR was used to analyze the gene expression induced by tryptamine (TAM) injection (10 ng per insect, after 4 h) in the fat body cells of locusts. The results showed that tryptamine could significantly induce the expression of *LmMao-a*, *LmMao-b*, *CYPs*, and immune genes (*stubble*, *cactus*, and *easter*) but suppress the expression of *defensin*. Data were presented as mean ± SEM. ** indicated P < 0.01. Bar, 100 μm. Blue: nucleus, red: DHE-stained cells (D), Immunoblot analysis was used to observe the AhR receptor translocation between the cytoplasm and nuclear space. Tryptamine (10 ng per insect) significantly induced AhR translocation 4 h after injection. GAPDH and histone H3 were used as the internal controls for loading

proteins of cytoplasmic and nuclear proteins, respectively. The band intensity was calculated by using Quantity One (Bio-rad, USA). The data were presented as mean ± SEM and analyzed by using one-way ANOVA. Different lowercase letters indicated significance at P < 0.05. (E) Fluorescent confocal microscopy imaging showed that the tryptamine inhibitor SR1 (86 pg, 860 pg, and 8.6 ng per insect) reduced the ROS production of tryptamine (10 ng per insect) in fat body cells. Bar, 100 μm. Blue: nucleus, red: DHE-stained cells. (F), qPCR was used to observe the expression of detoxification and immune genes affected by tryptamine (n = 6, different lowercase letters indicate significant difference at P < 0.05) after injecting different dosages (86, 860, and 8.6 μg/insect) of SR1. qPCR showed that SR1 suppressed the tryptamine-induced gene expression of *CYP*s, *stubble*, *cactus*, and *easter* genes in the fat body cells of locusts but induced the tryptamine-suppressed gene expression of *defensin*. The data were presented as mean ± SEM and calculated by one-way ANOVA in SPSS 20.0 software. TAM: tryptamine. (G) Moreover, dsRNA injection (20 μg/insect) of AhR effectively suppressed the *LmAhR* expression by qPCR analysis. (H) dsRNA injection successfully reduced the TAM induction of ROS production in fat body cells. Bar, 100 μm. Blue: nucleus, red: DHE-stained cells. TAM: tryptamine. Consequently, (I) the reduction of AhR expression by RNAi showed the same effects of SR1 on the mRNA level of detoxification and immune genes. All data were presented as mean ± SEM and analyzed by Student's t-test. **, P < 0.01. TAM: tryptamine. (J) Immunoblot analysis was used to observe the receptor translocation between the cytoplasm and nuclear space and detect the activation of tryptamine receptor AhR. The results showed that MAA and MAC infections induced AhR translocation. (K) Locusts were injected with tryptamine (100 pg/individual) before fungal infection to determine the toxicity of tryptamine. The results showed that tryptamine enhanced the lethality of fungi MAA and MAC to locusts. The survival curves of the insects were calculated by using Kaplan–Meier methods (P < 0.05).

that MAC infection activated higher levels of AhR than did MAA infection (Fig 3J). Moreover, the virulence of MAA and MAC were significantly enhanced (P < 0.05) by the administration of 100 pg tryptamine per insect (a sub-lethal dosage, $LD_{30}$) (Fig 3K). Overall, the effect of tryptamine on oxidative-stress induction and immune inhibition promoted fungal virulence.

## MAC genome lacks the *Mao* gene for tryptamine catabolism

Quantification of the tryptamine level in mycelia and hemolymph of locusts after fungal infections was conducted by using high-performance liquid chromatography (HPLC). The results showed that the tryptamine peak appeared at about 24.5 min in HPLC analysis. Importantly, the mycelia sample of MAC had a high level of tryptamine, whereas the mycelia sample of MAA displayed very little tryptamine (Fig 4A). The hemolymph of MAC-infected locusts had approximately a 1.8-fold higher level of tryptamine (266.90 ± 9.10 pg/μL) compared with MAA-infected locusts (152.68 ± 6.67 pg/μL) 4 days post-infection (P < 0.05) (Fig 4B). Subsequently, we performed HPLC quantification analysis of the tryptamine level in fungal mycelia and found that the level accumulated in MAC cells (35.59 ± 3.92 ng/mg) was 5.0-fold higher than that of MAA cells (6.57 ± 0.81 ng/mg) after 3 days of growth in a Sabouraud dextrose broth (SDB) medium. Similarly, the tryptamine level in MAC (85.07 ± 9.04 ng/mg) was 2.5-fold higher than in MAA (34.47 ± 1.94 ng/mg) after MAC culture in L-15 medium was amended with locust hemolymph (Fig 4C). We also observed an approximately 2.0-fold higher level of tryptamine in the culture filtrates of MAC than those of MAA (SDB: 52.77 ± 1.79 pg/μL of MAA, 94.32 ± 2.56 pg/μL of MAC; L15: 5.98 ± 0.13 pg/μL of MAA, 9.80 ± 0.42 pg/μL of MAC, P < 0.05) (Fig 4D).

After a genome-wide analysis of genes involved in tryptamine production or conversion in different *Metarhizium* species, we found that the mammalian-like *Mao* gene is absent in specialist fungi, including MAC and *Metarhizium album*. Phylogeny analysis revealed that three *Mao* paralogous genes *MrMao* are present in MAA: MAA03753 (*MrMao-1*, 30% identity to human MAO-B P27338), MAA08578 (*MrMao-2*, 28% identity to human MAO-B), and MAA10281 (*MrMao-3*, 27% identity to human MAO-B) (Fig 4E and S2 Fig). The schematic graph shows the key difference between MAA and MAC after investigation of the KEGG pathway of tryptamine catabolism pathway in *Metarhizium* genomes (Fig 4F). These results implied that MAC was able to accumulate a higher level of tryptamine because of the absence of *Mao*.

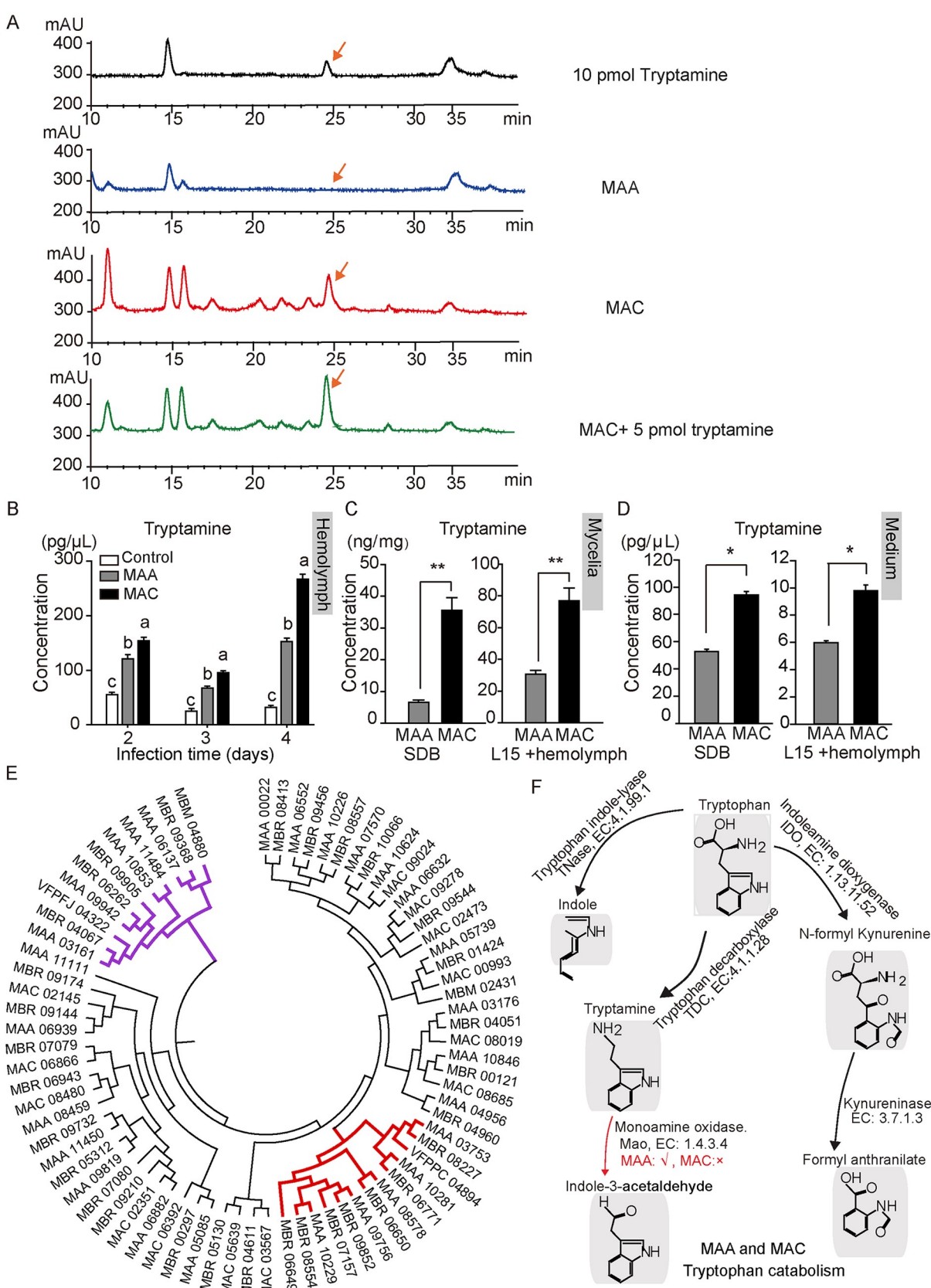

**Fig 4. MAC produced a higher level of tryptamine than did MAA because of the absence of *Mao* genes.** (A) HPLC analysis of tryptamine level in mycelia of MAA and MAC. The results showed that the mycelia of MAA presented little of tryptamine while samples of MAC presented a high level of tryptamine. The tryptamine peak appeared at about 24.5 min. (B) HPLC analysis of the tryptamine level in the hemolymph of locusts after being infected by MAA and MAC at different times. The results showed that the tryptamine level increased according to the time of infection, and MAC infection induced the highest level of tryptamine in locusts (n = 12, different lowercase letters indicate the significance, P < 0.05, one-way ANOVA for analysis). (C) After being cultured in SDB medium for 3 days or in L-15 medium with locust hemolymph for 6 days, the MAC mycelia showed a higher level of tryptamine than those of MAA. $n_{SDB}$ = 7, $n_{L15}$ = 18. Asterisks show the significance at P < 0.01. (D) HPLC analysis showed that the medium of MAC presented a higher level of tryptamine than that of MAA. (n = 10, asterisk indicated significance at P < 0.05 level). (E) Phylogeny analysis of protein sequences was used to classify the distribution of MAO proteins in *Metarhizium* species. Red lines show the mammal-like MAO; purple lines represent the FAD-dependent oxidases. The VFPPC_04894 (amine oxidase, *Pochonia chlamydosporia*) and VFPFJ_04322 (FAD-dependent oxidoreductase, *Purpureocillium lilacinum*) were used to indicate amine oxidase classification. The phylogeny tree was constructed via the neighbor-joining method by using MEGA 7.0 software. MAA: *Metarhizium robertsii*, MAC: *Metarhizium acridum*, MBR: *Metarhizium brunneum*, and MBM: *Marssonina brunnea*. (F) After analyzing the tryptophan catabolic pathway of MAA and MAC in the KEGG database, we found that MAC lost the *Mao* gene for tryptamine oxidation (EC: 1.4.3.4) in its genome, whereas MAA possessed three Mao (*MrMao*) genes.

## *MrMao-1* gene in *Metarhizium* controlled the tryptamine level in host

MAA was grown in SDB for 3 days, and qRT-PCR analysis was performed. The results showed that the expression level of the *MrMao-1* gene was approximately 100.0-fold higher than those of the *MrMao-2* and *MrMao-3* genes (Fig 5A). The three putative genes (*MrMao-1*, *2*, and *3*) were individually deleted in MAA to determine their roles in MAA function (Fig 5B). The deletion of *Mao* genes did not affect the growth of mutant strains of MAA (Fig 5C). However, the tryptamine levels in fungal cells and culture filtrates significantly increased more than 2.0-fold in the mutant strains of MrMao-1 (Δ*mao-1*) compared with the deletions of *MrMao-2* and *MrMao-3* (Fig 5D). Moreover, after 4 days of inoculation, the Δ*mao-1* strain produced 247.03 ± 25.71 pg/μL tryptamine in the hemolymph of locusts, approximately 1.6-fold higher than in wild-type (WT)-infected locusts (152.68 ± 5.82 pg/μL) and approximately 7.7-fold higher than in non-infected control locusts (32.11 ± 3.23 pg/μL) (Fig 5E).

## Mutant MAA showed higher virulence to insects than wild type

Only MAC had induced a higher level of cellular ROS accumulation in fat body cells 4 days after inoculation of fungal conidia compared with the mock control, as observed using a ROS-sensitive dye, dihydroethidium (DHE). Likewise, Δ*mao-1* strain induced the accumulation of a higher level of ROS in the fat bodies of locusts relative to the WT strain of MAA (Fig 6A). The locust bioassays consistently indicated that the virulence of the Δ*mao-1* strain was significantly increased (P = 0.027), as evidenced by the shorter life span of locusts by around 19.2% (34 hours) compared with that of the WT strains (Fig 6B and S3 Table).

As MAA has a broad host range of insects, we inspected the host range change of the Δ*mao-1* strain. Infected bioassays by Δ*mao-1* strain included mealworm (*Tenebrio molitor*), pea aphids (*Acyrthosiphon pisum*), mosquito (*Aedes aegypti*), silkworm (*Bombyx mori*), and American cockroach (*Periplaneta americana*). The results revealed that the Δ*mao-1* strain was still able to kill and mummify different phylogenic groups of insects but did not alter the host range change of MAA (Fig 6B). Specifically, the virulence of the Δ*mao-1* strain to mealworm and silkworm was significantly increased (P = 0.029 and P < 0.01, respectively) by shortening the life span of the mealworm and silkworm by approximately 18.3% (38 hours) and 31.28%, respectively, compared with that of the WT strains. The virulence of Δ*mao-1* strain to pea aphids, mosquito, and cockroach were all significantly increased (P < 0.01, P < 0.01, and P = 0.05, respectively) by shortening these insects' life span by around 21.3% (26 hours), 14.2% (21 hours), and 15.8% (55 hours), respectively, compared with the treatment with WT strain of MAA (S3 Table). Thus, tryptamine accumulation improved the virulence of MAC against insects and activated the AhR of the host to induce ROS stress (Fig 6C).

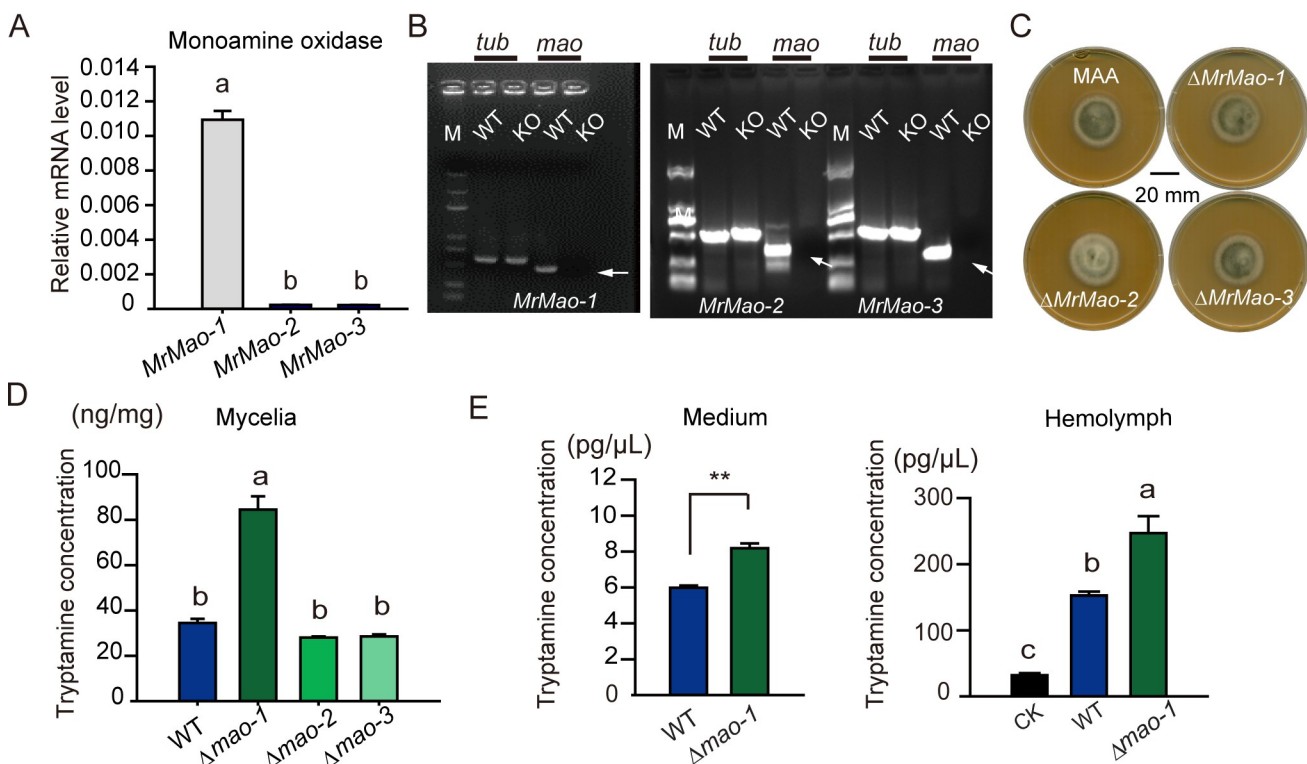

**Fig 5. Deletion of *MrMAO* genes in MAA significantly promoted the tryptamine concentration in mutants.** (A) qRT-PCR analysis showed that *MrMao-1* presented the highest level of mRNA expression in the *MrMao* gene family. (B) By introducing the pDHt-MAO-bar plasmid, we knocked out the *MrMao* genes in MAA. RT-PCR results showed that *MrMao* genes (*MrMao-1*, *MrMao-2*, and *MrMao-3*) were successfully knocked out from the MAA genome. M: DNA ladder D2000; WT: wild type of MAA; KO: MAA samples of knockout *MrMao*, *tub*: tubulin gene. White arrow indicates the target bands of Δ*mao* strain samples. (C) The colony morphology of mutant and wild-type MAA strains showed little difference in the PDA medium. (D) HPLC analysis of the tryptamine level in the hyphae cultured in the L-15 medium with locust hemolymph supplement. The results showed that knocking out *MrMao-1* increased the tryptamine level, but *MrMao-2* and *MrMao-3* did not affect the tryptamine level in MAA (n = 8–14, different lowercase letters indicated a significant difference at P < 0.05, WT: wild type). (E) The left panel showed the extracellular concentrations of tryptamine in the MAA strain of WT and Δ*mao-1*. The Δ*mao-1* strain showed higher levels of tryptamine in the mycelia and medium than did the WT strain. The data were presented as mean ± SEM, Student's t-test. Asterisks showed the significance at P < 0.01. WT: wild type. The right panel showed that the Δ*mao-1* strain infection induced a higher level of tryptamine in the hemolymph of locust than did the WT strain by HPLC analysis. CK: mock control (n = 10, different lowercase letters indicated significance at P < 0.05).

## Discussion

In the present study, the two *Metarhizium* spp., MAA and MAC, had significantly different virulence and insecticidal mechanisms. Generalist MAA uses destruxins to kill the insect host, whereas specialist MAC accumulates the high concentration of tryptamine to poison insects. The absence of *Mao*, which controls the catabolism of tryptamine, in the genome of MAC leads to the accumulation of tryptamine. Hence, deleting *MrMao-1* in the genome of MAA promotes the tryptamine level. In addition, the presence or absence of *Mao* in the genome of *Metarhizium* spp. leads to the differences in virulence between MAA and MAC. Δ*mao-1*, a gene-modified strain of MAA, showed high virulence to many species of insects and did not affect host ranges including *Orthoptera*, *Diptera*, *Coleoptera*, *Hemiptera*, *Lepidoptera* and *Blattaria*. The findings shows that the high insecticidal virulence of MAC relied on gene deletion instead of gene insertion. Because the deletion of enzyme genes in catabolic pathway leads to the accumulation of toxic metabolites for insects, the approaches for promoting insecticidal virulence of fungi can be used to develop highly efficient mycoinsecticides without introducing any exogenous gene. Therefore, a new method was proposed in this study to modify existing

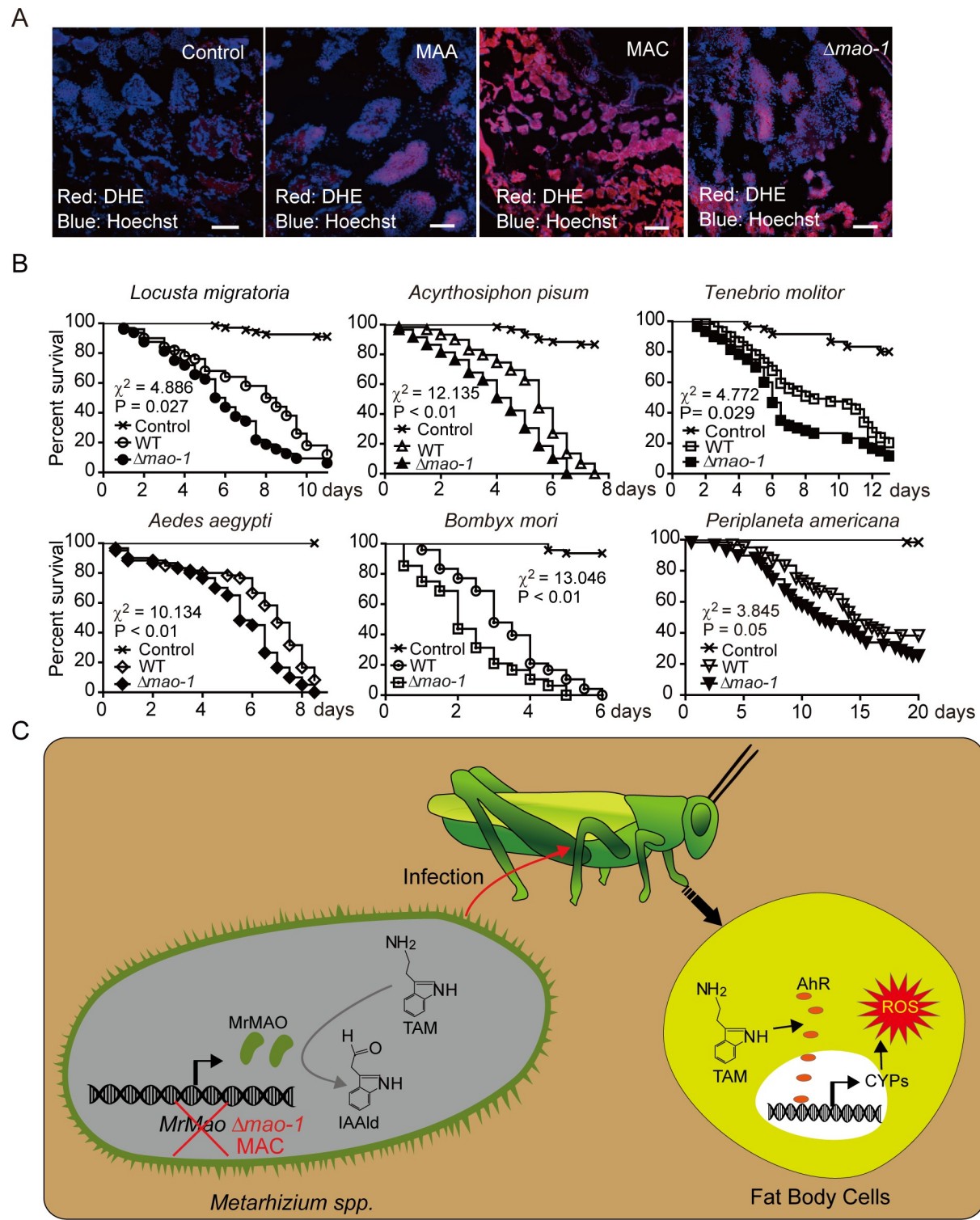

**Fig 6. Mutant MAA distinctly promoted virulence against many insect species.** (A) Fluorescent confocal microscopy imaging of MAC and Δ*mao-1* showed that both had increased levels of tryptamine that induced more ROS production in locust fat body cells. Red cells indicated that DHE was degraded by ROS for staining the DNA of cells. Blue cells indicated Hoechst 33342 staining. (B) Δ*mao-1* strain showed a higher efficacy of killing six insect species than the WT strain. The survival curves of insect species were calculated by using Kaplan–Meier methods. P < 0.05. (C) Schematic graph showed that the mechanism of *MrMao-1* gene controls the tryptamine level to regulate the damage effects on the host. TAM: tryptamine, IAAld: Indole-3-acetaldehyde.

mycoinsecticides that can easily promote the virulence of *Metarhizium* and prevent environmental and health concerns of public communities.

We confirmed that tryptamine was the virulence factor of MAC that directly resulted in the death of locusts because MAC was not able to produce the virulence factor destruxin. Tryptamine in plants can regulate the feeding behavior and reproduction of herbivorous insects [24–26]. Tryptamine, serving as a precursor for the growth and defense hormones in plants, is directly derived from tryptophan and is the most expensive amino acid only synthesized in plants and microbes [27,28]. In addition, synthesized derivatives of tryptamine are toxic to insects [19]. MAA has a broad range of hosts, whereas MAC only has the specific hosts limited in orthopteran insects. Although narrow host ranges of MAC reduced its efficiency for harvesting tryptophan from insects, the tryptamine accumulation of MAC ensured its superiority of competence with MAA in promoting plants growth. Tryptamine can also increase the efficacy of MAC to kill insects and hasten the recycling of tryptophan from insects to plants [29]. Notably, the number of PKS and PKS-like genes that synthesize secondary metabolites in MAC are half of those in MAA [7]. The smaller number suggests that MAC may rely on simple metabolites to improve its virulence against insects owing to the biosynthesis cost. MAC can produce other cytotoxic antibiotic metabolites [11], however the pathogenesis of such metabolites in contributing to the virulence of MAC still needs to be investigated.

The insecticidal mechanism of tryptamine in locusts is involved in activating AhR and increasing the ROS stress in fat body cells, thus leading to immune suppression in locusts. Our study showed that AhR activation induced a high level of ROS stress from tryptamine detoxification in locust fat body cells, because AhR responds to tryptamine [30], initiates the expression of detoxification genes, and suppresses the immune responses of hosts [23]. Interestingly, crowded locusts highly presented inhibitors of ROS production including *serpin*s, *pacifastin*, and *redoxin*s in respond to MAA infection in comparison to solitary locusts [16], suggesting that ROS production affected the tolerance to fungal pathogens in locusts. Indeed, ROS has harmful effects on life span of organisms because it caused the damage to cells [31,32]. Besides, the high level of serine proteases induced by tryptamine in locusts can enhance ROS production by increasing the susceptibility of the prophenoloxidase cascades [33,34]. Tryptamine suppresses the immune response by enhancing *cactus* gene expression to inhibit the downstream of antimicrobial peptide (defensin) production. Thus, the immune suppression may increase the opportunistic infection of other microbes, which increases the mortality of insect hosts. On the basis of these findings, tryptamine catabolism is the key mechanism that systematically regulates host responsiveness to *Metarhizium* spp. and provided the infection opportunity of other microorganisms.

Although MAA and MAC species possess the entire enzyme systems for tryptophan catabolism, including TDC, tryptophan indole-lyase, and indoleamine dioxygenase, the degradation of tryptamine at downstream of tryptophan catabolism is blocked in MAC due to the absence of *Mao* genes. Indeed, the gene numbers in the MAC genome are lesser (approximately 2,000 genes) than those in the MAA [2,11]. The simplicity of the MAC genome probably leads to the attenuated rate of monoamine catabolism. Because plants preferred the organic nitrogen resources [35,36], MAC is possibly better adapted to plants as symbiont due to the accumulation of amines than MAA that catabolized amines to ammonia. On the contrary, the production of ammonia benefits MAA for both induction and activation of fungal subtilisin proteases in the process of insect infection [37]. The important role of tryptamine catabolism in the different virulence of MAA and MAC implied that the manipulation of enzymes of tryptophan synthesis (especially TDC enzymes) in MAA will produce an extremely high virulence of mycoinsecticides by elevating the tryptamine level.

We deleted the *MrMao-1* gene in MAA according to the heuristic of the specialist MAC genome and obtained the fungal mutant Δ*mao-1*, which promoted the lethality of the fungus against many insect species. The Δ*mao-1* strain of MAA possesses the additional virulence factor tryptamine besides destruxins. Aside from promoting virulence, Δ*mao-1* still killed six species of insects, which belong to very different evolutionary groups. Despite fungi presenting distinct metabolites to adapt to different insect tissues [38], we observed that tryptamine accumulation does not alter the host ranges. Although high virulent MAA can infect beneficial insect including silkworm, application method such as accurate dispersal onto locations of pest outbreak can limit the off-target to other insects. Moreover, this approach of modifying tryptamine catabolism in other narrow host range of *Metarhizium* species such as *Metarhizium pingshaense* can avoid the risk of infecting non-target hosts [39]. Further, manipulating the fungal recognition on insect particular cuticle components can improve the host specification and is essential to guarantee the safety of other mycoinsecticides in future.

Unlike the introduction of exogenous genes to improve the efficacy of mycoinsecticides, the manipulation of tryptamine catabolism avoids the risk of the transgene of fungus in the biosphere. Fungus with tryptamine accumulation can also be used as bio-fertilizer to promote crop growth [40]. Thus, a novel approach was proposed to design efficient mycoinsecticides for environmental safety and nitrogen recycling of insects in agro-ecosystems.

## Materials and methods

### Locust inbreed

The locusts (*L. migratoria*) used in this experiment were collected from one locust population (Hebei Huanghua, China) and reared under standard conditions in the lab [41]. They were cultured alone in metal boxes ($10 \times 10 \times 25$ cm$^3$) supplied with charcoal-filtered compressed air. The locust colonies were maintained under a 14 h light/10 h dark cycle at $30°C \pm 2°C$ and fed fresh wheat seedlings. The male adults were used in our experiments three days after molting.

### Fungal strains and culture conditions

Two *Metarhizium* species, MAA (ARSEF 23) and MAC (CQMa 102), were used in this study [10]. Fungal cultures were maintained on potato dextrose agar (Sigma, USA) at $26°C$ for 2 weeks, and conidia were harvested to prepare spore suspensions ($1 \times 10^6$ conidia/mL). For liquid culture incubation, an aliquot (100 μL) of the spore suspension from each species was inoculated into a 100 mL flask containing 25 mL SDB (BD Difco) [11] or cultured in an L-15 medium (Invitrogen, USA) amended with locust hemolymph (10 μg/mL).

### Transcriptome analysis

The responses of locusts to MAA and MAC infection was investigated using three treatments of samples for RNA-seq: the control, fat body samples after 4 days of MAA infection, and fat body samples after 4 days of MAC infection. Each treatment had three biological replicates. After the total RNA extraction and DNase I treatment, magnetic beads with Oligo (dT)$_{15}$ were used to isolate mRNA. The mRNA was mixed with the fragmentation buffer and fragmented into short fragments. Then, cDNA was synthesized using the mRNA fragments as templates. Fragments with a length of 200 bp were purified and resolved with an EB buffer for end reparation and single nucleotide A (adenine) addition. Afterward, the fragments were connected with index sequences. The suitable fragments were selected as templates for PCR amplification. During the quality control steps, the Agilent 2100 Bioanalyzer and ABI StepOnePlus real-time

PCR system were used in the quantification and qualification of the sample libraries. Finally, the libraries were sequenced using Illumina HiSeq 4000. The differentially expressed transcripts were analyzed using DEGseq and edgeR software. The sequenced data were uploaded to the Sequence Read Archive (SRA) database (SRA accession: SRP145755).

## RNA isolation and qRT-PCR analysis

All samples were directly frozen in liquid nitrogen until RNA and protein sample preparation. Total RNA was extracted by using a TRIzol reagent (Invitrogen, USA). The cDNA was synthesized from 1.5 μg of total RNA using the MLV reverse II system (Promega, USA). qRT-PCR was conducted using a Roche Light Cycler 480 system (Roche, Mannheim, Germany) with SYBR green master mix (Roche, Mannheim, Germany). qPCR was initiated with a 10 min incubation at 95˚C, followed by 45 cycles each of 10 s at 95˚C, 20 s at 58˚C, and 20 s at 72˚C. The primers designed based on PRIMER 5.0 are listed in S4 Table.

## Immunoblot analysis

Cytoplasmic and nuclear fractions of locust fat body cells were prepared by using a nuclear and cytoplasmic protein extraction kit (Beyotime, Jiangsu, China). Proteins were separated by 12% SDS-PAGE. The proteins were transferred to the polyvinylidene fluoride membranes by using a transfer buffer. Next, the membranes were blocked by 5% (wt/vol) skimmed milk for 4 h at room temperature. The membranes were incubated in a blocking buffer with a primary antibody at the following concentrations: anti-AhR (1:200 dilution) and anti-GAPDH (1:5,000) at 4˚C overnight [16]. The AhR antibody was prepared by expressing C-terminal protein (accession number: MH393891, ORF: 1387–1911 bp) in *Escherichia coli*. Glyceraldehyde 3-phosphatase (GAPDH) and histone H3 (Sigma, USA) were used as cytoplasmic and nuclear protein controls, respectively. Secondary antibodies (goat anti-rabbit IgG and goat anti-mouse IgG, HRP conjugated, 1:5,000, CoWin, Beijing, China) in the blocking buffer were used to detect the primary antibodies for 1 h at room temperature. The protein signals of the immunoblot membrane were detected using the SuperSignal West Femto Substrate Trial Kit (Invitrogen, USA).

## ROS detection

The locusts were quickly frozen by liquid nitrogen and then embedded with O.C.T. compound (O.C.T. is abbreviation of Optimal Cutting Temperature, Sakura Finetek, USA). The embedded locusts were stored at −20˚C overnight for cryosection. Next, 10 μm cryosections of different treatment locusts were prepared. These sections were stained with 2 μM DHE (Sigma, USA) for 1 h in a humidified box and then fixed in 4% (wt/vol) paraformaldehyde for half an hour. Finally, the samples were washed thrice with phosphate-buffered saline [42]. Hoechst 33342 (Invitrogen, USA) was used to stain the cell nucleus. The images were captured on an LSM 710 confocal fluorescence microscope (Zeiss, Germany).

## HPLC analysis of tryptamine

Fungus cultured in either SDB or L-15 medium amended with locust hemolymph for 3 or 6 days were collected for further investigation. Liquid medium was filtrated for collecting fungus, and then the mycelia and culture filtrates were freeze dried for further experiments. Subsequently, the hypha samples were ground sufficiently, and 1 mg of powder was collected for protein removal by adding 100 μL of 0.1 M perchloric acid. After centrifuging at 5,200 × g for 10 min, the supernatant was neutralized by 1 M $Na_2CO_3$ to pH 6.0. Then, the supernatant was

collected into a sample bottle for HPLC detection [43]. Every 1 mL medium was treated in a vacuum freeze dryer and added with 50 μL 0.1 M perchloric acid to remove proteins. After centrifuging, the samples were neutralized by 50 μL 1 M $Na_2CO_3$. The hemolymph was immediately collected and centrifuged at 500 × g for 10 min at 4˚C to collect the supernatant and detect tryptamine in the locusts' hemolymph. The cell pellet was then discarded. The procedures for protein removal were the same as those of the hypha samples. All prepared samples were detected by Agilent 1100 equipped with a G1315A fluorescence detector. The column used for HPLC was Agilent Eclipse XDB-C18 (5 μm, 4.6 × 250 $mm^2$). The tryptamine standard (Sigma, USA) was dissolved in 0.1 M acetic acid. The derivatization reagent 1,2-phthalic dicarboxaldehyde was prepared according to a previous research [43]. The HPLC conditions were as follows: solvent A is a mixture of a solution (0.05 M acetate buffer/tetrahydrofuran [96/4]) and methanol at a ratio of 60:40, and solvent B is 100% methanol. The pH of solvent A was adjusted to 6.0. The gradient elution of target molecules was as follows: solvent A (in %): 75.00 (0 min), 75.00 (8 min), 66.67 (12 min), 50.00 (25 min), 0 (30 min), 66.67 (35 min), and 75.00 (40 min); solvent B (in %): 25.00 (0 min), 25.00 (8 min), 33.33 (12 min), 50.00 (25 min), 100 (30 min), 33.33 (35 min), and 25.00 (40 min).

## Gene deletion in fungus

Individual deletion of the three putative *Mrmao* genes (*MrMao-1*, *MrMao-2*, and *MrMao-3*) was performed through homologous recombination as described previously [44]. Briefly, two primer pairs were used to amplify the 5′- and 3′-flanking regions of the target gene. The PCR products were digested with the restriction enzymes SmaI and SpeI. The treated fragments were inserted into the corresponding sites of the binary vector pDHt-bar (conferring resistance against ammonium glufosinate) to generate the plasmid for *Agrobacterium tumefaciens*-mediated transformation. The transformants were verified via PCR and RT-PCR analyses. The β-tubulin gene (MAA_02081) was used as an internal control during the detection of target gene expression.

## Drug administration

Tryptamine was injected into locusts 24 h after SR1 injection, and the samples were collected after 4 h. The different dosages of tryptamine (0.1, 1, 10, 100, and 1,000 ng per insects) were used to determine $LD_{30}$. In addition, 10 ng tryptamine was injected to each locust after 1, 2, and 4 h to detect the superoxide in the fat body. The AhR antagonist SR1 was injected as 86 pg, 860 pg, and 8.6 ng into the locust for 24 h. Then, the dosage of 10 ng per insect for tryptamine injection was used to conduct the bioassay.

## Knockdown of AhR by dsRNA

The dsRNA of the AhR (NCBI accession number: MH393891, ORF: 1,230–1,875 bp) and a negative control gene (green fluorescent protein) were prepared by using the T7 RiboMAX Express RNAi system (Promega, Madison, USA) according to the manufacturer's instruction. The insect was injected with 20 μg dsRNA for 24 h. The RNAi effect on the expression of target genes was detected by using qRT-PCR. The primers used are shown in S4 Table.

## Insect bioassays

Three days after molting, the locusts were individually inoculated under pronotum with 2 μL peanut oil (Sigma, USA) containing $1 × 10^6$ conidia/mL of MAC or MAA [7]. The locusts were maintained in individual containers under a 14 h light/10 h dark cycle at 30˚C with

enough food. The mortality of insects was recorded every 12 h. The survival curves were calculated by using Kaplan–Meier methods. SPSS 20.0 software was used in this analysis [16].

Different insects were used for bioassays to compare the virulence difference between strains WT and Δ*mao-1*. Each treatment used 60 mealworms (*T. molitor*, 5th instar, approximately 2 cm). The conidial suspension was prepared in peanut oil at $1 \times 10^6$ conidia/mL. The insects were given 1 μL oil suspension [45]. The control group was treated with peanut oil without conidia. Then, each insect was placed into a 12-well culture plate with bran which was moved into an incubator set at 26˚C. Insect mortality was recorded at 12 h intervals.

Adult insects with similar sizes were selected and transferred onto a detached leaf in a 15 cm-diameter petri dish to determine the differential virulence against pea aphids (*A. pisum*). The leaf was wrapped with a sterile cotton ball to maintain its freshness. Nymphs emerged after 24 h, and were removed from the leaf. The nymphs developed into adults within 7 days and were used in the following bioassay [46]. The conidial suspension was prepared in a 0.05% sterile Tween-20 aqueous solution at $1 \times 10^8$ conidia/mL. Fresh pea leaves were soaked in the conidial suspension for 1 min and wrapped with a sterile cotton ball to maintain freshness in a $35 \times 10 \text{ mm}^2$ petri dish. The pea leaves were then placed in a $90 \times 16 \text{ mm}^2$ petri dish with 30 adult aphids. Subsequently, they were maintained under a regime of 21˚C ± 1˚C and 16 h of light and 8 h of dark photoperiod. Each of the control, WT of MAA, and Δ*mao-1* treatments had 60 adults. Moreover, the leaves were changed after 2 days. Insect mortality was recorded at 12 h intervals.

We used the methods described in previous studies [47] to test the virulence against mosquitoes. Mosquitoes (*A. aegypti*) were held at 27˚C ± 1˚C with 80% ± 5% RH. The conidia were diluted with 0.05% Tween-20 to obtain a concentration of $1 \times 10^8$ conidia/mL. Peanut oil was added to this solvent to generate 10% oil formulations. Afterward, 6 mL of the formulations was placed evenly over a $15 \times 4 \text{ cm}^2$ piece of filter paper using a pipette. The impregnated papers were left to dry at 70% RH for 48 h and then placed inside a glass tube (height 14 cm, diameter 2.5 cm). The 4-day-old mosquitoes were placed inside this tube with access to 6% glucose solution placed in freshly soaked cotton [47]. The control was treated the same but without conidia. Each treatment used 60 mosquitoes. Insect mortality was recorded at 12 h intervals.

The infection procedures of silkworm (*B. mori*, 5th instar) were the same as those of mealworm (*T. molitor*). Each group had 48 larvae in a clean plastic case with holes on the lid. Silkworms and mealworms were provided with fresh *Folium mori* every day. Insect mortality was recorded at 12 h intervals.

Cockroaches (*P. americana*) were placed in a cold chamber at 4˚C for half an hour. All cockroaches were inactivated and infected with 2 μL oil suspension ($1 \times 10^8$ conidia/mL) under the pronotum. They were then placed in the dark under 27˚C and provided enough food and water. Insect mortality was recorded at 12 h intervals.

### Statistical analysis

SPSS 20.0 (SPSS Inc., USA) was used to analyze all statistical data. Independent-sample Student's t-tests were used to compare the difference in gene expression between treatments. The survival curves of the six insect species were calculated by using Kaplan–Meier methods. The threshold of P-value was adjusted by Bonferroni correction.

### Supporting information

**S1 Fig. GO enrichment analysis of differentially expressed genes from the direct comparison of MAA- and MAC-infected locusts.** (A) Class of biological process in MAA- and MAC-

infected locusts. (B) Class of molecular function in MAA- and MAC-infected locusts.
(TIF)

**S2 Fig. Computer simulation of fungus Mao protein families to human MAO.** A 3-D structure of MrMAO proteins was constructed by Protein Homology/analogY Recognition Engine V 2.0 (Phyre2). The protein structures of MrMAO were found to be highly similar to human MAOB (PDB number: 1S3B). The root mean square deviations of MrMAO-1, MrMAO-2, and MrMAO-3 calculated by FATCAT simulation (http://fatcat.burnham.org/fatcat/) were 2.38, 1.46, and 2.76, respectively. The red structure is the human MAOB, and the gray structure is MrMAO.
(TIF)

**S1 Table. Top DEGs uniquely induced by MAC infection.**
(DOCX)

**S2 Table. Calculation of lethal dosage of tryptamine to locusts by Probit methods.**
(DOCX)

**S3 Table. LT50 data of six insect species after infection of WT and *Δmao* strain of MAA.**
(DOCX)

**S4 Table. All PCR primers used in the experiments.**
(DOCX)

# Acknowledgments

We are grateful to Drs. Cui Feng, Zou Zhen, and Huang Lihua for the supplies of the additional insects *Acyrthosiphon pisum*, *Aedes aegypti*, and *Bombyx mori*, respectively.

# Author Contributions

**Conceptualization:** Xiwen Tong, Yundan Wang, Le Kang.

**Data curation:** Xiwen Tong, Yundan Wang, Pengcheng Yang, Chengshu Wang.

**Formal analysis:** Xiwen Tong, Yundan Wang, Chengshu Wang.

**Funding acquisition:** Yundan Wang, Le Kang.

**Investigation:** Yundan Wang, Chengshu Wang, Le Kang.

**Methodology:** Xiwen Tong, Yundan Wang.

**Project administration:** Yundan Wang.

**Resources:** Yundan Wang, Chengshu Wang.

**Software:** Pengcheng Yang.

**Supervision:** Yundan Wang, Le Kang.

**Validation:** Yundan Wang, Le Kang.

**Visualization:** Yundan Wang.

**Writing – original draft:** Xiwen Tong, Yundan Wang, Chengshu Wang, Le Kang.

**Writing – review & editing:** Yundan Wang, Chengshu Wang, Le Kang.

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
