## [Decision Letter · Decision Letter 0]

19 Dec 2019

Dear Dr Kang,

Thank you very much for submitting your Research Article entitled 'Tryptamine accumulation caused by deletion of MrMao-1 in Metarhizum genome significantly promotes insecticidal virulence' to PLOS Genetics. Your manuscript was fully evaluated at the editorial level and by independent peer reviewers. The reviewers appreciated the attention to an important topic but identified some aspects of the manuscript that should be improved.

We therefore ask you to modify the manuscript according to the review recommendations before we can consider your manuscript for acceptance. Your revisions should address the specific points made by each reviewer.

[LINK]

Yours sincerely,

Jens Rolff

Guest Editor

PLOS Genetics

Gregory P. Copenhaver

Editor-in-Chief

PLOS Genetics

Pest species such as locusts cause major losses of harvest. One important control measure for pest insects and diseases vectors are Mycoinsecticides, fungi that are professional insect pathogens. How and why some these fungi kill their insect hosts and whether this can be harnessed for control measures is studied in this article.

The authors compare the virulence between the host-specialist fungus Metarhizium acridumn (MAC) and the generalist Metarhizium robertsii (MAA). They find that the specialist is more virulent and that this virulence is caused by tryptamine. A genomic comparison shows that MAC are devoid of an important that catabolizes tryptamine. They then study the immune response of the host and can show that tryptamine elicits a damaging reactive oxygen response in the host and also suppresses the expression of defensins. They also recreate the virulent tryptamine producing genotype in MAA and find the same virulence as in MAC.

Overall, this is a very complete story of understanding the virulence of the fungus and the killing mechanism of the host. The authors also point out how that insight can be utilized for controls.

The paper in general is easy to follow and well structured. The paper would nevertheless benefit from polishing the language. Reviewer 2 provides a number of points that should be closely adhered to, including the access to the data.

For readers not familiar with pest control it would be useful to have one or two sentences on the importance of insect pests or in particular locusts.

In the discussion I feel it should addressed that a more virulent entomopathogen would not only infect the target host, but basically all insects. How is this problem going to be resolved? See perhaps the comments on Lovett et al 2019 Science.

Some minor comments.

Line 105-107: statement unclear

Line 779-781 is unclear to me and does not seem to match the main text.

Line 325-327 meaning unclear.

Reviewer's Responses to Questions

**Comments to the Authors:**

Reviewer #1: Well written and well thought out experimental design. The only thing I would suggest is to check the grammar and syntax throughout the manuscript.

Reviewer #2: Article Under Review: Tryptamine accumulation caused by deletion of MrMao-1 in Metarhizium genome significantly promotes insecticidal virulence

Summary:

In this study, the authors carefully characterize the high amount of tryptamine production by the specialist insect pathogenic fungus Metarhizium acridum, as compared to the generalist pathogen Metarhizium robertsii. They further convincingly link the production of tryptamine higher virulence by the specialist toward locusts, which is mediated by ROS accumulation via LmAhR. Through comparative genomics and transcriptomics, they discovered the tryptamine accumulation in the specialist is due to its lack of a monoamine oxidase gene required for tryptamine catabolism. This finding led to the production of a strain of M. robertsii wherein they deleted this gene. This mutant recapitulated the increased virulence and increased ROS seen in M. acridum, but irrespective of insect host. It seems they have uncovered a tryptamine-mediated path to enhanced virulence in pathogenic fungi with applicability in a wide range of insects. They rightly highlight that this method would not require transgenesis, as it could be accomplished through gene deletion.

Comments:

Ln 32: Throughout, is it more correct to say “inducing” instead of “regulating”?

Ln 46: I suggest referring to “MAA” as “MRO” throughout for consistency.

Ln 144: Please indicate which solvent you used to make your spore suspension.

Ln 152-154: Is it more appropriate to say “up-regulated” instead of “expressed”, “triggered” and “activated”?

Ln 155-156: Does this section of the Venn Diagram really show this? Did you limit your MAA vs. MAC comparisons to only MAC genes? Please make this more clear.

Ln 185-188: These accession numbers do not pull up anything on NCBI, please check these.

Ln 252: Fascinating!

Ln 327: I recommend explicitly stating that this will be advantageous from a regulatory perspective.

Ln 342-344: What does this sentence mean?

Ln 345: Add a citation to justify “hasten the recycling of tryptophan from insects to plants”.

Ln 355: This sentence seems to switch the order of AhR and ROS stress. Consider rewording.

Ln 379: Your diagram suggests TDC, not Mao, is involved in the conversion tryptamine from tryptophan.

Ln 382-388: Why would the generalist be better adapted to insects than the specialist? Also, why would the specialist be better adapted to plants? Generalists tend to be better plant symbionts than specialists. This section needs to be removed or rethought.

Ln 391: This statement about HGT is unfounded speculation, particularly considering how closely related Pochonia chlamydosporia and Metarhizium species are. Please remove.

Ln 308-409: Please add a citation to back up this statement.

Fig 1: Consider reporting percent survival on the y-axis instead of log rank throughout. Also, consider using colorblind-friendly coloration.

Suggested Revisions:

Ln 2: change “promotes” to “enhances”

Ln 37: add space after “insects.”

Ln 42: remove “traditional”

Ln 43: change “is a fungus that” to “fungi”

Ln 44: change “lives” to “live”; change “produces” to “can produce”

Ln 50: add comma after “destruxins”

Ln 63: change “these” to “the”

Ln 64: change “safe” to “their safety”

Ln 70: change “its” to “their”; change “it” to “they”; change “differentiates” to “differentiate”

Ln 79: add comma after “hosts”

Ln 82: change “germling” to “germlings”

Ln 85: add “cuticular” before “penetration”; change “has” to “have”

Ln 86: change “its” to “their”; remove “the” before “virulence”

Ln 87: remove “of its pathogens”

Ln 94: change “in charge of” to “required for”

Ln 96: remove “enzymes”

Ln 108: change “while” to “during”; swap “MAA” and “locusts”

Ln 125: add “the” after “that”

Ln 127: add “the” after “whereas”

Ln 132: change “multi-copies” to “multiple copies”

Ln 135: add “factor” after “virulence”

Ln 151: change “each time of” to “their respective”

Ln 175: change “part” to “components”

Ln 188: remove comma

Ln 203: remove “the”

Ln 216: change “nuclei” to “nucleus”

Ln 235: change “missed” to “lacks the”

Ln 236: remove “a”; capitalize “Quantification”; remove “analysis”; remove “the” after “in”

Ln 335: change “destruxins” to “destruxin”

Ln 337: change “plant” to “plants”

Ln 346: change “plant” to “plants”

Ln 352: change “confirmed” to “investigated”

Ln 364: change “damages in” to “damage to”

Ln 373: add period after “spp”

Ln 408: replace “horizontal gene” with “transgene”

Ln 420: remove “after”

Ln 421: change “of” to “after”

Ln 482: write O.C.T. in full

Ln 532: change “Drug” to “Tryptamine”

Ln 607: change “P value” to “p-value”

**Have all data underlying the figures and results presented in the manuscript been provided?**

Reviewer #1: Yes

Reviewer #2: Yes

PLOS authors have the option to publish the peer review history of their article (what does this mean?). If published, this will include your full peer review and any attached files.

Reviewer #1: Yes: MJBidochka

Reviewer #2: No

---

## [Editor Report · Decision Letter 1]

29 Jan 2020

Dear Dr Kang,

Thank you very much for submitting your Research Article entitled 'Tryptamine accumulation caused by deletion of MrMao-1 in Metarhizum genome significantly enhances insecticidal virulence' to PLOS Genetics. Your manuscript was fully evaluated at the editorial level.

We therefore ask you to modify the manuscript according to the minor recommendations by the associate editor (see below) before we can consider your manuscript for acceptance. Your revisions should address the specific points made by each reviewer.

In addition we ask that you: Upload a Striking Image with a corresponding caption to accompany your manuscript if one is available (either a new image or an existing one from within your manuscript). If this image is judged to be suitable, it may be featured on our website. Images should ideally be high resolution, eye-catching, single panel square images. For examples, please browse our archive. If your image is from someone other than yourself, please ensure that the artist has read and agreed to the terms and conditions of the Creative Commons Attribution License. Note: we cannot publish copyrighted images.

[LINK]

Yours sincerely,

Jens Rolff

Guest Editor

PLOS Genetics

Gregory P. Copenhaver

Editor-in-Chief

PLOS Genetics

The authors have done a great job. The paper is very close. There are a few places that should be corrected, which I think makes the paper easier to read:

Line 88 I assume you mean specificity here, not specification

Line 90-91 sentence unclear: is it meant that 15 % of all putative genes are virulence genes?

Line 92 Do you mean ‘virulence genes’ here

Line 198 replace than with compared to

Lines 206-207: Rephrase. By comparing….., it was found that….

Line 214. …. To be between 1,585. – 20795 (put Cis in brackets)

Line 225: insect ‘in insects’ after AhR

Line 330 I think you mean ‘differences in’ here, rather than ‘diverse’?

Line 334 showS

Line 334 Relied instead of relayed

Line 334 gene not genes

Line 348 is expansive really meant, or expensive or widespread?

---

## [Editor Report · Decision Letter 2]

17 Feb 2020

Dear Dr Kang,

We are pleased to inform you that your manuscript entitled "Tryptamine accumulation caused by deletion of MrMao-1 in Metarhizum genome significantly enhances insecticidal virulence" has been editorially accepted for publication in PLOS Genetics. Congratulations!

Yours sincerely,

Jens Rolff

Guest Editor

PLOS Genetics

Gregory P. Copenhaver

Editor-in-Chief

PLOS Genetics

Comments from the reviewers (if applicable):

**Data Deposition**

http://datadryad.org/submit?journalID=pgenetics&manu=PGENETICS-D-19-01917R2

**Press Queries**

---

## [Editor Report · Acceptance letter]

19 Mar 2020

PGENETICS-D-19-01917R2 

Tryptamine accumulation caused by deletion of MrMao-1 in Metarhizium genome significantly enhances insecticidal virulence 

Dear Dr Kang, 

We are pleased to inform you that your manuscript entitled "Tryptamine accumulation caused by deletion of MrMao-1 in Metarhizium genome significantly enhances insecticidal virulence " has been formally accepted for publication in PLOS Genetics! Your manuscript is now with our production department and you will be notified of the publication date in due course.

With kind regards,

Jason Norris

PLOS Genetics

On behalf of:
